# Maritime Robotics and Autonomous Systems Operations: Exploring Pathways for Overcoming International Techno-Regulatory Data Barriers

**Tafsir Matin Johansson** [1,*], **Dimitrios Dalaklis** [2] and **Aspasia Pastra** [1]

[1] World Maritime University-Sasakawa Global Ocean Institute, World Maritime University, 21118 Malmö, Sweden; asp@wmu.se

[2] Maritime Safety and Environmental Administration, World Maritime University, 21118 Malmö, Sweden; dd@wmu.se

\* Correspondence: tm@wmu.se; Tel.: +46-764-088-238

**Abstract:** The current regulatory landscape that applies to maritime service robotics, aptly termed as robotics and autonomous systems (RAS), is quite complex. When it comes to patents, there are multifarious considerations in relation to vessel survey, inspection, and maintenance processes under national and international law. Adherence is challenging, given that the traditional delivery methods are viewed as unsafe, strenuous, and laborious. Service robotics, namely micro aerial vehicles (MAVs) or drones, magnetic-wheeled crawlers (crawlers), and remotely operated vehicles (ROVs), function by relying on the architecture of the Internet of Robotic Things. The aforementioned are being introduced as time-saving apparatuses, accompanied by the promise to acquire concrete and sufficient data for the identification of vessel structural weaknesses with the highest level of accuracy to facilitate decision-making processes upon which temporary and permanent measures are contingent. Nonetheless, a noticeable critical issue associated with RAS effective deployment revolves around non-personal data governance, which comprises the main analytical focus of this research effort. The impetus behind this study stems from the need to enquire whether "data" provisions within the realm of international technological regulatory (techno-regulatory) framework is sufficient, well organized, and harmonized so that there are no current or future conflicts with promulgated theoretical dimensions of data that drive all subject matter-oriented actions. As is noted from the relevant expository research, the challenges are many. Engineering RAS to perfection is not the end-all and be-all. Collateral impediments must be avoided. A safety net needs to be devised to protect non-personal data. The results here indicate that established data decision dimensions call for data security and protection, as well as a consideration of ownership and liability details. An analysis of the state-of-the-art and the comparative results assert that the abovementioned remain neglected in the current international setting. The findings reveal specific data barriers within the existing international framework. The ways forward include strategic actions to remove data barriers towards overall efficacy of maritime RAS operations. The overall findings indicate that an effective transition to RAS operations requires optimizing the international regulatory framework for opening the pathways for effective RAS operations. Conclusions were drawn based on the premise that policy reform is inevitable in order to push the RAS agenda forward before the emanation of 6G and the era of the Internet of Everything, with harmonization and further standardization being very high priority issues.

**Keywords:** robotic and autonomous systems; drones; crawlers; remotely operated vehicles; Internet of Robotic Things; data governance; user interface; international standards

## 1. Introduction

Digital data, commonly referred to as data, have a ubiquitous influence in the contemporary information-intensive age, as vast quantities of data are created every second, if

not microsecond. The dynamic benefits of data acquisition and subsequent usage have led end-users to the general understanding that data have an unfeigned nexus with decision-making processes, especially ones that concern business-as-usual.

Over the years, the term data has found itself being defined in a myriad of ways. It is further observed that each individual definition of data can be characterized as being discipline-oriented. Whether natural science, economy, or law, scholars have put forth respective definitions from the prism, and to the extent, the term data has interacted with the subject matter of respective disciplines [1]. The quintessential definition of data, however, is found in the common lexicon, i.e., the Cambridge English Dictionary which defines the term as "information, especially facts or numbers, collected to be examined and considered and used to help decision-making, or information in an electronic form that can be stored and used by a computer" [2]. All-embracing in content and ambit, the above definition found in the common lexicon reflects the essence of the varying definitions, to all intents, construction, and purposes, propounded by various subject matter experts, e.g., Senn (1982), Clare and Loucopoulos (1987), and Avison and Fitzgerald (1995) [3–5].

Despite the abundance of discipline-led definitions, experts view modern-day dependency on data with a positive outlook. What began as a mere byproduct of business processes, is now incontrovertibly a valuable asset [6,7]. Similar to land-based industrial counterparts, this so-called dependency on data was observed in the maritime and ocean domain, for example, in marine science concentrated on species distribution modelling, geospatial data in maritime boundary delimitation, submarine communications, nautical charts, and marine scientific research of areas beyond national jurisdiction [8–10]. In short, data are invaluable and reside at the crux of work undertaken by stakeholders that participate in effecting a sustainable transformation in relation to maritime and ocean governance.

There is, of course, the shipping industry's constant reliance on "timely and accurate data to feed its logistical plan" [11]. To take but one example, the voyage data recorder (VDR), which is fitted into passenger ships and other ships of 3000 gross tonnage and above, is considered an important compliance tool in line with the International Maritime Organization's (IMO) International Convention for the Safety of Life at Sea). Not only has VDR facilitated forensic investigations in post incidents and casualties in the past, but also the data gathered by investigators have remained of interest to stakeholders, including academics and researchers that support "green operations" through performance efficiency to ameliorate overall safety while reducing operational costs. Research on data continues with efforts focusing on exploring the means and methods to process "big data" and "data analytics" from an ethical–socio–technical setting [12,13]. In this journey of exploration, the acquisition and translation of "big data" to promote safety culture is a promising direction taken by the shipping industry, following the illustrative development/precedent in the "Big Data Traffic Information Service" system by the automotive company Toyota in 2013 [14].

Technology is undoubtedly a catalyst of change [15]. This statement itself is well grounded in facts, especially considering the manner in which RAS are currently being integrated and adapted into the operational framework of business activities to obtain data. The shipping industry is certainly no exception. Clean and green shipping, one of the clearest manifestations of RAS integration, comes off as a timely response in mitigating the adverse effects of shipping on climate change where data analysis has an invaluable purpose. Second to this crucial aspect is the integration of RAS to carry out statutory and classification tasks, e.g., annual survey, periodic survey, and special survey coupled with damage repairs, satisfied via conventional methods with a view to obtain real-time data for observation of structural weaknesses, thus altogether contributing to maintaining international vessel performance standards [13]. Once fully integrated, RAS would eventually replace traditional human-led survey, inspection, and maintenance. The end-objective of all RAS-driven operations is to gain optimum accuracy in the process of data acquisition during inspection and to improve post-inspection decisions based on data acquired. For

example, outer hull is an area prone to damages caused by biofouling. Determining biofouling status for maintaining vessel speed and increasing maneuverability is contingent on accurate data in relation to a number of integral structures. In principle, the processes are governed by international laws and requirements, adopted at the national level and adhered to by flag state authorities [13].

The paradigm shift of RAS reliance on shipping has already begun [15]. In 2020, Bureau Veritas recorded the test results of using drones for close-up inspections and thickness measurements, setting a milestone in this RAS-deployment continuum [16]. Drones, also known as unmanned aerial vehicles (UAVs), are aircrafts that "can be remotely controlled or programmed to fly a predetermined route using information on a specific asset's condition to target known areas of concern" [17]. The other two existing RAS applications include crawlers and ROVs. Very briefly, the former is designed to "crawl along a structure . . . to operate on a vertical surface or hull structures in air or underwater" [17]. The latter is exclusively used in underwater operations to "collect visual data, perform Nondestructive Testing (NDT), and measure plate thickness in difficult-to-reach areas" [17].

RAS-interventions today are very data-centric, as they are translated into activities such as data acquisition, data transfer, data storage, and finally data analyses, which remain the principal foci. Although RAS applications—as they exist today—are best described as systems with "human-in-the-loop", their relevant capabilities are promising, leading to the projection that drones, crawlers, and ROVs will achieve full autonomy in the near future [13,15]. As RAS advances towards full autonomy, there is a clear need to spear through the current data governance and corresponding management practices to assess whether procedural details, rules, and requirements follow a water-tight approach securing much needed precautions that would otherwise give rise to complications from the get-go stage.

In furtherance of the foregoing, this paper discusses the results gathered from the European Union Horizon 2020 funded project titled Autonomous Robotic Inspection and Maintenance on Ship Hulls (BUGWRIGHT2) while addressing two pragmatic questions: what are the thorny issues that could invoke data liability in an RAS-led vessel survey and inspection operation, and what are the pathways through which those issues could be mitigated? The motivation behind seeking answers to the above emanates from the current vacuum that exists in the regulatory setting that could call on liability implications that are purely legal in nature. It is important that service robotics engineers, as they move from "human-in-the-loop" technology to "full autonomy" technology, remain aware that technology and law reside in the same continuum and are therefore interrelated. Technology should progress, but in that process, it must be ensured that such progress does not breach or violate any contemporary legal provisions that are globally sensitive in nature. Data acquisition is one subject matter of crucial importance to both engineers as well as global, regional, and national regulators. Simply put, a solution is required before RAS is in mass deployment and before the topic of data protection reaches the contentious stage.

With a view to finding a solution to the questions posed above, this article focuses on identifying the principal data barriers existing in the international techno-regulatory framework (which is followed by ship owners, classification societies, as well as the operational folks engaged under the title of service providers), as well as potential solutions for consideration based on the theoretical construct of data governance. International norms and standards set by the International Association of Classification Societies (IACS) are the ones that regulate all RAS vessel survey activities and are sporadically revised as needed to ensure that the provisions are fit-for-purpose. Based on the current setting as it exists, it is correct to state that shipowners lack a techno-regulatory safety net to protect the non-personal data linked to their asset. The growing needs to eliminate liability drawbacks from such an absence are being discussed in different platforms at the European Union level, including the European Commission funded project, ROBotics technology for Inspection of Ships (ROBINS).

With the above in mind, the paper commences with an examination of the characteristic peculiarities of the two main classes of data, and this evidently supports the placement of RAS-acquired data under the category of non-personal data, and further allows the authors to explore non-personal data decision domains and building blocks. This is followed by an analytical highlight of the types of stakeholders involved in the non-personal data business model with specific reference to human and non-human actors. Subsequently, this paper provides an expository review of the state-of-the-art international legal framework that maritime RAS falls within. This is done with reference to current international data-related rules and requirements, highlighting the entities typically engaged in vessel survey and inspection operations involving RAS data acquisition, analysis, and validation.

Additionally, an attempt is made to provide an incisive examination of the architecture of Internet of Robotic Things (IoRT) in the context of maritime RAS and through the prism of data. Thereafter, a comparative examination is conducted to highlight best practices offered by selected IMO-recognized organizations, referred to as classification societies that are proactively engaged in regulating vessel survey and inspection processes, including the main subject matter of this article, i.e., data. The major drawbacks of the current international framework are extracted to underline noteworthy challenges before carving out ways forward that provide a first-hand synoptic insight into the regulatory blueprint developed by researchers of the World Maritime University (WMU) under the ongoing BUGWRIGHT2 project. The paper concludes with takeaway notes synthesizing important points gathered from previous discussions, and ones that will allow for digital serenity as maritime RAS engineering progresses.

## 2. Materials and Methods

To satisfy the aims and objectives of this paper, a combination of the doctrinal and comparative methods was employed. The doctrinal methodology is basically concerned with doctrinal research, which in turn is a combination of "legal theory research" and "expository research" [18]. Given that the paper focuses on technologies engineered to collect data, "legal theory research" (which underpins the detailed study of legal doctrines and principles, jurisprudence, and legal philosophy) remains outside the scope of the work [18].

To promote the results of this article, the authors relied on the technological regulatory (techno-regulatory) international framework (as a part of the "legal theory research") that governs all RAS affairs under IMO's rule of reference. Reliance on the international framework was unavoidable given that they are followed by the RAS community at large, including engineers, manufacturers, technology specialists of RAS, human–machine interface psychologists, shipowners, flag states, as well as service providers. The primary method is therefore expository research that is concerned with the study of technology-based legislation, international treaties, and scholarly literature that touch upon data-related laws, rules, and requirements with respect to RAS—applications of which are in force. This methodology is used to analyze the extant law (i.e., *de lege lata*), pointing out its drawbacks and deficiencies. The subject must be thoroughly understood in order to determine what the law should be in the future (i.e., de lege ferenda). Needless to say, this approach highlights the continuum of past, present, and future in terms of the progress of the law aimed at technologies.

It is important to note that expository research mainly comprises detailed the examination of legal texts that govern the usage of RAS, including international rules and requirements found in legislative material and international legal instruments. In the world of the science–policy interface, these are known as "black-letter law". The exposition of legal texts in the research process includes non-treaty instruments, relevant scholarly literature such as textbooks, academic and professional journals containing legal, management and governance opinions and expert commentaries, standard form international contracts, and the like.

The comparative methodology—essentially comprising comparative research—is used to compare best practices offered. It extends to the examination of law and legal practices in light of current developments, the object of which is to carry out a comparative analysis. The information so gleaned can lead to obtaining an insight into best international practices. Comparative legal research is deemed to be an important tool that can help the researcher to review and juxtapose best practices through rational comparisons. Within the framework of comparative research, interviews were also conducted with relevant stakeholders of the data management process, including an executive of a service supplier company, a legal expert, an IT shipping consultant and a drone team of a maritime technical service company.

It is envisaged that the final outcome of the examination and critical analysis of techno-law carried out through the use of the doctrinal and comparative methods of legal research leads to strategic proposals and pragmatic alternatives being presented by the authors. These should take into account the specific aspects of RAS data governance and management that need changing or introduced. The proposals for ongoing regulatory development should be aimed at making future laws more coherent, precise, and practical (de lege feranda), thereby providing adequate security and protection to beneficiaries.

## 3. Results

### 3.1. Theoretical Construct: Dissecting the Dimensions of Data

Despite contemporary high usage, the term "data governance" continues to remain elusive [19,20]. With reference to the definitions implemented by the Data Management Association (DAMA), the term "data governance" is viewed as "the allocation of authority and control and shared decision making over the management of data assets" [21]. With this definition in mind, secondary sources have carved out the end-objectives of data governance: enhancing data quality, data value, and reducing data-related cost and risk [22–25].

At the outset, it is important to note that the principal concept behind data governance brings to the forefront the need for the comprehension of the inherent dichotomy that lies between personal and non-personal data, since any given data could be a combination of different datasets and transformed into "personal data" in cases where there is processing power and data availability (Mattoo and Meltzer, 2018; Chatzimichali and Chyrostomou, 2019) [26,27]. The definition of personal data, which is of practical significance and far from being of theoretical interest, is intrinsically related to data that directly or indirectly relates to an identified or identifiable natural person (Finck and Pallas, 2020) [28].

A concrete definition of "personal data" is found in the EU's Regulation 2016/679 on the General Data Protection Regulation (GDRP). According to Article 4(1) GDPR, the term personal data is defined as "any information relating to an identified or identifiable natural person ('data subject'); an identifiable natural person is one who can be identified, directly or indirectly, in particular by reference to an identifier such as a name, an identification number, location data, an online identifier or to one or more factors specific to the physical, physiological, genetic, mental, economic, cultural or social identity of that natural person" [29] (Article 4.1). Non-personal data, on the other hand, venture outside the scope of identifiable natural person, as they relate to industrial and anonymized raw machine-generated data [30].

Legal and business-environment facets pertaining to non-personal data are regulated by the European Union's (EU) Regulation 2018/1807, as outlined in the framework for the free flow of non-personal data in the European Union—an exemplary governance tool that aims to achieve data-driven growth and innovation between the EU Member States. In the above Regulation, non-personal data are conceptualized as "data other than personal data", as defined in Paragraph 9 of the GDPR and include anonymized datasets used for big data analytics and data on maintenance needs for industrial machines [31]. The Regulation moves unjustified barriers for the free movement of non-personal data and applies to EU natural or legal persons that provide data processing services (e.g., collection, recording,

storage, use, disclosure by transmission, dissemination of data), giving individuals and organizations the opportunity to collect and disseminate non-personal data and to use data centers or cloud services located within the EU while protecting individual privacy.

In the context of terminology differentiation, it is also important to note the distinction between data governance and data management, since "governance" is the de facto overarching term within which data management resides. Data governance is related to the upper-level planning and the decisions about the allocation of responsibilities, access, control, and use of data, as opposed to data management, which is linked to the implementation and monitoring of governance-related decisions [32]. Recently published literature indicated that "data management" encompasses the essential sequence of the following activities: collection, storage, processing, using, sharing, and destroying of data [33]. Conversely, data governance ensures appropriate data management and provides the relevant processes and structures for formally managing information and for protecting data as a strategic asset [33–35].

There is also a narrower focus that merits further observation, namely that organizational and technical perspectives should be jointly included in a proper data governance framework [36]. Recent efforts include the work by Abraham et al. [25], who set out to conduct an in-depth literature review that eloquently identified the following data decision domains and building blocks with regards to non-personal data:

Governance mechanisms:

- Identification of the data owner/data manager/data consumer
- allocation of responsibilities
- decision-making authority
- organizational policies and standards
- coordination with the stakeholders;

Application of governance mechanisms on the organizational scope, data scope, and domain scope:

- Quality and integrity of project-related organizational data
- data governance between firms
- machine-generated data
- privacy requirements
- data architecture
- data lifecycle
- data storage;

Antecedents of data governance:

- Internal and external factors that precede data governance practices, such as organizational culture and industry;

Outcomes of data governance:

- Measurement of data governance effectiveness on organizational strategy and business performance [25].

Most of today's data governance programs address goals in two or three data decision domains. Within the settings of the above cited data governance framework, several interrelated decision domains remain dormant: (a) data architecture; (b) data quality; (c) data security and data access; (d) data lifecycle; (e) metadata; and (f) data storage and infrastructure [25,32,37].

The first decision domain, i.e., "data architecture", is a term that refers to the blueprint that aligns data assets with the organizational strategy and sets designs to meet data requirements [21]. In short, the architectural policies and standards denote how data are processed, stored, distributed, and used. Data architecture helps to facilitate the flow of data within an organizational ecosystem by developing interrelated and bidirectional data pipelines. The second domain known as "data quality" focuses on the ability of data to fulfill their intended use through precision, timeliness, completeness, and credibility [32].

In this regard, specific organizational metrics are utilized to safeguard data quality. As for the decision domain that is explicitly tied with safeguarding measures, "data security" as the third domain touches on many issues, including privacy, confidentiality, intellectual property policies, data access, and third-party access [38]. The third domain does not ignore the possibility that effective organizational security mechanisms and tools could help protect data from unauthorized access and corruption throughout their lifecycle.

Similar to biorhythm, the lifecycle of data under the fourth decision domain titled "data lifecycle" captures the intricate timespan of the life of data from its generation and maintenance, up to the very point of their deletion. Data lifecycle represents the approach of archiving and retiring big data until it no longer makes sense to retain them, leading to the assumption that overmanaging information could lead to the waste of capital resources [39]. In terms of data generation, "metadata", whether created manually or automatically, is the fifth category conceptualized as "data-about-data" or as "information stored in IT tools that improves both the business and technical understanding of data and data-related assets" [38]. Metadata are stored in a location different from the original data and are generally invisible to the end-user. There are different types of data that can be considered as "metadata": title, author, keywords, permissions, geolocation of an image, copyright information, date and instruction for the users, among others. Finally, "data storage and infrastructure", the sixth decision domain, is about maintaining or archiving digital data on different types of media for usage by computer devices [40]. Technological storage options are ample whereby technology provides economical and reliable solutions for storage arrangements.

### 3.2. Carving Out the Stakeholders of Non-Personal Data

The need to identify the types of stakeholders that directly or indirectly affect or are directly or indirectly impacted by data decision domains (described in the previous section) with regards to non-personal data is unavoidable in discussions related to data governance. As the first step, one must be cognizant of the key human and non-human actors both within and outside the organization [41]. The interactions between multiple actors have their roots in the actor–network theory (ANT), which is quite often utilized to analyze systems in the fields of social sciences, international relations, and information technology [42].

ANT explores the relational ties within a network in which knowledge is perceived as the product of a network of heterogeneous actors [42]. ANT examines the construction and transformation of heterogeneous networks and collaborative designs, taking social and technical aspects into account. Each network has a generalized symmetry, and each actor has an equal role to play. Therefore, ANT sets the framework for alliance among humans, machines, tools, computers, and other non-human agents. Actions are a conglomerate of many agency groupings that must be slowly untangled [43,44]. An actor–network relationship is never stabilized, owing to the fact that agents are not isolated from each other and are constantly involved in the process of actions and reactions [42,45].

Human actors in a data framework that concerns "machines" are the stakeholders who are involved in and are affected by data-based decisions and actions. The stakeholders that belong to the data governance framework are the participants that are involved in information sharing throughout the data supply chain management [46].Potential stakeholders range from manufacturers, software developers, service providers, business clients, information architects, data governance practitioners, metadata analysts, operations staff, and end-users [47]. Then there are non-human actors that interact with the above: policies and regulations, market demands, economic circumstances, semiotics, technological tools and infrastructure, inter alia [41].

It would seem that the notion of a "governance network" that introduces human and non-human actors' interrelations is of paramount importance in order to mark out the stakeholders that belong to the service robotics non-personal data framework. In doing so, one will come across an important detail: human agents are not unique when it

comes to shaping and influencing reality, and for this reason, non-human actors' role in an actor–network domain should not be diminished [44]. Any given system contains diverse entities whereby each entity is a sum of other smaller actors. Inevitably, international standards embedded within the scope of policies and regulations, viewed as a non-human factor, influence the manner in which tasks are completed, covering the essentials that are required. In doing so, international standards cement the close and enduring partnership between and among the stakeholders in the fulfilment of non-personal data acquisition, analysis, and validation objectives.

### 3.3. RAS and the Internet of Robotic Things: Through the Prism of Data

In its Recommendation titled ITU-T Y.2060, the International Telecommunication Union (ITU) describes the Internet of Things (IoT) as an all-embracing international infrastructure for the informatics and communication society (International Telecommunication Union, 2012; Yousif, 2018) [48,49]. Consequently, IoRT comes forth as a concept that amalgamates IoT and cloud robotics, where cloud robotics is a combination of robotics and cloud computing, with the latter opening the digital doorway for sharing, processing, and storing interoperable information [49]. For maritime RAS, this corresponds to close-up information on vessel structures and thickness measurements of suspect areas.

The fabric of the IoRT architecture anticipates communication of information via five interdependent layers. Localized movements with the help of actuators and the conversion of signals or stimuli (light, motion, sound, and heat) to electrical domains via a wireless sensor network (WSN) form a part of the "hardware layer" that leverages information to the "network layer", which encompasses cellular connectivity, short and medium-range communication technologies, Worldwide Interoperability for Microwave Access (WiMAX), Z-Wave, ZigBee, and low power wide area network (e.g., LoRa) [50]. Within the architecture of IoRT, the "Internet layer" plays a central role, in so far as it supports lightweight information processing, inter alia, through transport protocols including data distributed services (DDS) in the robotics system for carrying out a number of principal tasks, listed as follows: "publish/subscribe messaging, multicast support, real-time instant messaging, packet switched networking, alternative to Transport Control Protocol, disseminating of networked embed system, providing privacy to datagram protocol, message queuing for middleware environment, lightweight local automation, and directly addressing publish/subscribe based communication for real-time embedded systems" [50]. The most invaluable layer of the architecture is the "infrastructure layer", which in turn comprises five distinct and interrelated components, namely a robotic cloud platform, M2M2A cloud platform support, IoT business cloud services, big data services, and an IoT cloud robotics infrastructure [50]. User interaction and interface through monitoring and visualization is realized in the last IoRT architecture element known as the "application layer"—the technical potentials of which are expanding [49,50]. Finally, the aforementioned individual but interconnected layers facilitate data acquisition from the physical world, which are fed to the application layer for observation. The advancements through machine learning (ML), a branch of artificial intelligence, are seen to have revolutionized the IoRT architecture with fast and smooth fault diagnosis, making "ships safer, easier to use and more efficient" [51,52].

Turning to maritime RAS for vessel survey, inspection, and maintenance, it was observed that the key parameters in the implementation and acceptance of RAS platforms in shipping industry are related to operational safety [15]. The so-called operational safety concerns ease of operation during intervention by "human-in-the-loop" during data acquisition, communication and data management, data security, energy autonomy, and power source. Although communication and data management tasks today are conducted using both paper and digital systems, data integrity is nevertheless ensured to a certain extent through the introduction of some form of a blockchain system in the data management process [15]. As the focus remains on integrating RAS into the current manual system, data security and the effectiveness of data collection, data processing, and distribution

of analysis outputs (e.g., defects autodetection) need to be demonstrated, ensured, and proven in an effective manner. For each of the three types of maritime RAS, the data communication system in maintaining the data decision dimension needs to be thoroughly revisited in light of user interface (see Table 1). Otherwise, RAS platforms are likely to lack trustworthiness among the stakeholders of the business model. Software developers need to provide details of international standards used and apply those standards to satisfy the data security requirements. For a clear understanding of the different stages involved in RAS in survey and inspection, a diagram (see Figure 1) is presented to familiarize the reader with a real live scenario.

**Table 1.** State of autonomy, class, and user interface of maritime drones, crawlers, and ROVs.

| Maritime Drones: Aviation/Air | | | |
|---|---|---|---|
| **Titles in usage** | **State of autonomy** | **Classes** | **User interface** |
| micro aerial vehicle, unmanned aerial vehicle (UAV), uncrewed aerial vehicle, remote inspection technology, remote inspection vehicle, multidrone system, multi-UAV, drone | Supervised autonomy (machine–human); remotely controlled | A1 (<250 g to <500 g): fly over people but not assemblies; A2 (<2 kg): fly close to people; A3 (<25 kg): fly far from people; CO (<250 g): can fly in subcategory A1 and A3; C1 (<900 g): can fly in subcategory A1 and A3; C2 (<4 kg): can fly in subcategory A1 and A2; C3 and C4 (<25 kg): can fly in subcategory A3). * **Note:** European Union Aviation Safety requires completion of training and passing of exams defined by the national authority for operating drones under category A1, A2, A3, C1, C2, C3, and C4 [53]. | High-performance data processing task can be achieved using computer graphical user interface (GUI) or natural user interface (NUI). GUI comprises a control panel, parameter viewer, camera viewer, and requested behaviors viewer. On the other hand, NUI types include visual body interaction, visual marker interaction, hand gesture interaction, and speech command interaction [54] |
| **Crawlers: Climbers on Structures (Predominantly Above Water):** | | | |
| **Titles in usage** | **State of autonomy** | **Classes** | **User interface** |
| Magnetic crawler, vertical surface magnetic crawler robot, remote inspection technology, remote inspection vehicle, magnetic utility crawler, magnetic crawler solutions, crawler | Supervised autonomy (machine–human); remotely controlled | Vacuum adhesion type: pneumatic climbing robot, articulated-limb crawler using rubber suction cups; magnetic adhesion type: electromagnetic wheeled crawlers; propeller type: see [55] | High-performance data processing task can be achieved using a 3D tracking concept and robot operating system. Specific location information is transmitted by tracking unit and visuals are communicated by built-in cameras [56]. |
| **ROVs: Underwater (Liquid Environment):** | | | |
| **Titles in usage** | **State of autonomy** | **Classes** | **User interface** |
| ROV, remotely operated underwater vehicle submersible, autonomous underwater vehicle, remote inspection technology, remote inspection vehicle | Supervised autonomy, supervised autonomy (machine–human), remotely controlled using cable and tether | Observation class (up to 100 kg, 300 m typical depth rating): DC voltage, 110/220 V DC/AC voltage); mid-sized class (from 100 kg to 1000 kg, 2000 m typical depth rating): 440/480 V, AC voltage; work-class (>1000 kg, >3000 m typical depth rating): 440/480V [57]. | The electronic architecture is fitted with computers—one is located at the surface and is connected to a monitor through which the user can view images taken by the camera and other sensors, and the second is located in the ROV's electronic contained. Both computers control the thruster and the underwater arm to read and process the sensor signal whereby all tasks are performed using data acquisition cards and National Instruments hardware. Communication is materialized via Ethernet protocol [58]. |

Sources: Adapted by authors from [53–58].

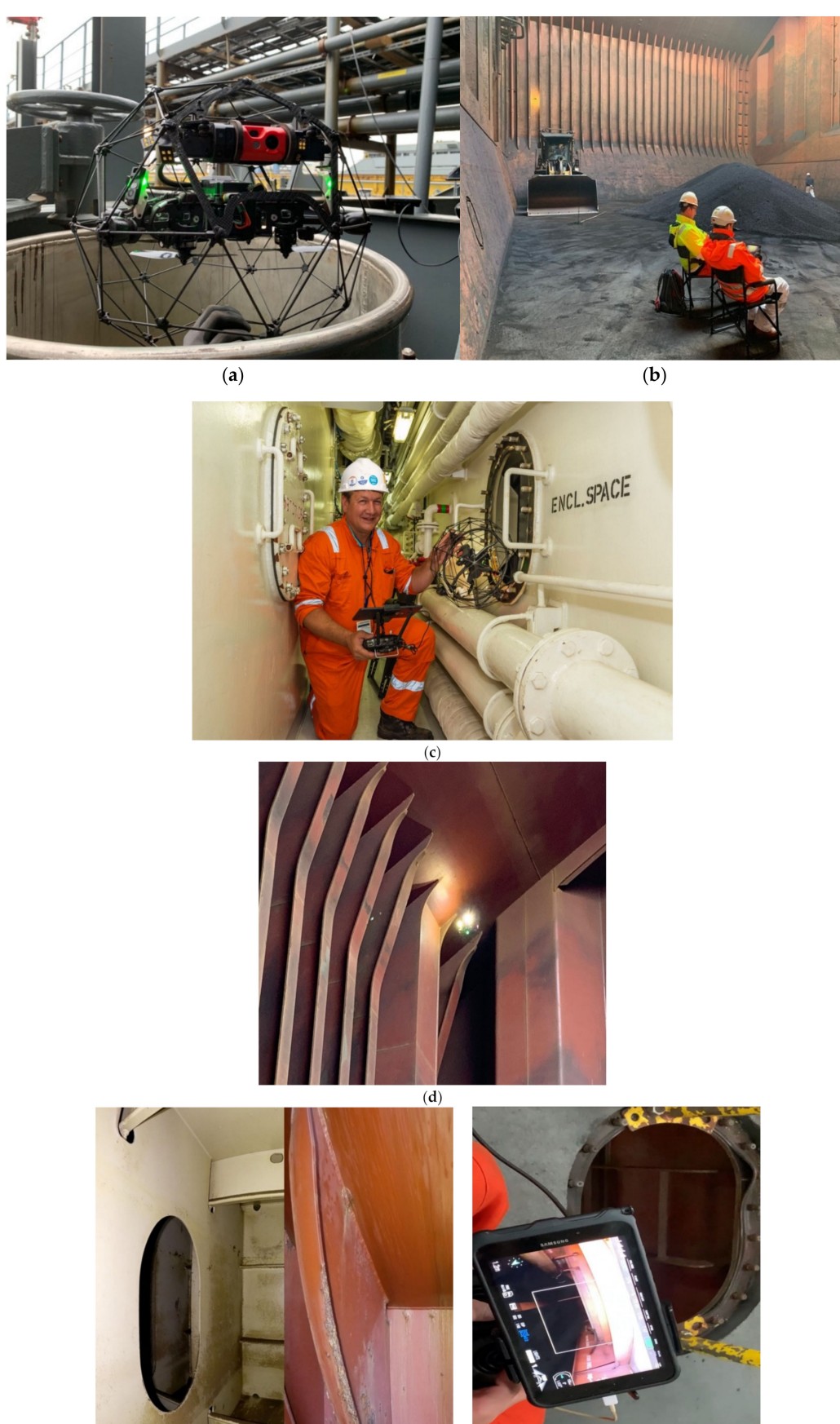

**Figure 1.** *Cont.*

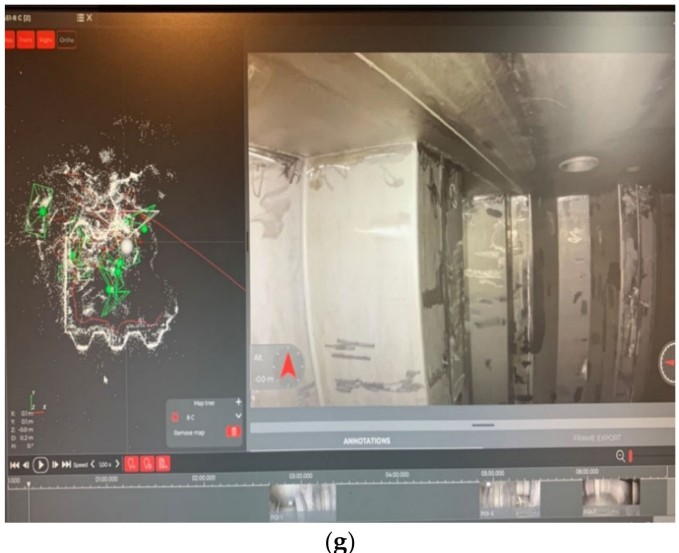

(**g**)

**Figure 1.** Overview of the operational sequences considering a drone as a case-study: a typical RAS calibration situation (**a**) and supervised-control during classification survey (**b**); a drone before "accessing" an enclosed space (**c**); drone during survey and inspection (**d**); visual image captured through drone (**left**) (**e**,**f**); data acquisition (**g**). Data can be live-streamed through video conference to facilitate remote surveys if required.

### 3.4. International Maritime Data Governance Standards

The data obtained and used via RAS clearly belong to the non-personal data domain, given that they concern a vessel and not a natural person. Usage of RAS for gathering non-personal data in shipping, as promulgated in the texts of IACS, stems from two main objectives: (1) to inspect the biofouling status of a ship and (2) to address elevated risks in other structural areas that could result in safety and environmental concerns [59,60]. For statutory surveys, the guidelines in IMO's "Guidelines for the Control and Management of Ship's Biofouling to Minimize the Transfer of Invasive Aquatic Species" issued in 2011 provides two options for conducting in-water inspection: human divers and remotely operated vehicles [60]. In other words, conducting inspection via ROV, i.e., a form of remote inspection technology (RIT), is acceptable and could lead to the issuance of statutory certificates by flag states [13].

In this discussion, the role of classification societies cannot be ignored. More than fifty individual classification society members have been quite influential actors within the shipping industry as far as the history of global maritime trade is concerned. Consequently, classification societies play a crucial role in meeting standards, with the focus recently shifting towards goal-based standards (GBS) through the development of standard rules and regulations in direct conformity with United Nations Convention on the Law of the Sea of 1982 (UNCLOS). It is not a coincidence that according to the official IMO website:

> "Goal-based standards (GBS) are high-level standards and procedures that are to be met through regulations, rules and standards for ships. GBS are comprised of at least one goal, functional requirement(s) associated with that goal, and verification of conformity that rules/regulations meet the functional requirements including goals. In order to meet the goals and functional requirements, classification societies acting as recognized organizations (ROs) and/or national Administrations will develop rules and regulations accordingly. These detailed requirements become a part of a GBS framework when they have been verified, by independent auditors and/or appropriate IMO organs, as conforming to the GBS".

The title under which classification societies operate in the process of developing GBS is known as Recognized Organizations (RO), a title spread across a significant number of IMO conventions covering environmental and safety facets. Of the many international

organizations that contribute to the creation of global maritime technology-based regulatory standards on classification and statutory rules, the IACS is viewed as the leading non-profit international body. Although the International Organization for Standardization has since 1947 has played the traditional role of developing technological standards for robotics and robotic products, the IACS's parallel efforts to date is observed as being the leader in the creation of a techno-regulatory environment for the maritime industry. In the evolution continuum, it is noted that the IACS has come a long way, in so far as its promulgated standards cover more than 90% of the global cargo carrying tonnage. In short, the IACS governs all policy matters that govern the usage of service robotics in the maritime and ocean industry [60]. Some scholars such as Lindøe and Baram purported that standard(s) are private industry driven and serve as an external non-governmental force that supports the national regulatory regime [61] (see Figure 2). Standards containing classification rules developed by the IACS are certainly no exception [60]. Classification surveys following classification rules lead to the issuance of classification certificates, which is an attestation of compliance [60].

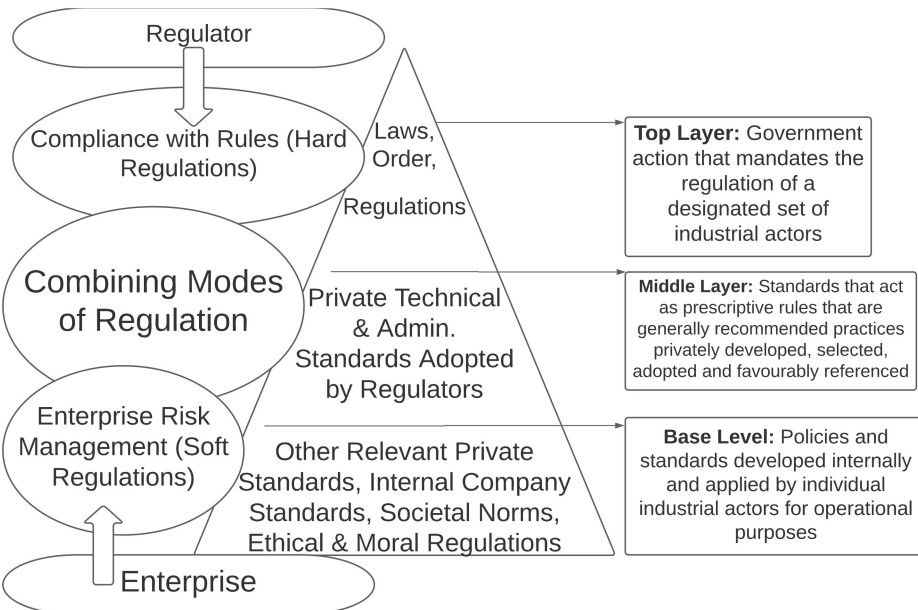

**Figure 2.** Standards in the regulatory governance regime. Source: Adapted by authors from [61].

Currently, the state-of-the-art survey and certification provisions reside among the various Unified Requirements (UR) developed by the IACS, such as UR Z3, UR Z7.1, UR Z7.2, UR Z10.1, and UR Z10.2. Notwithstanding the plethora of rules and requirements, the IACS has duly considered the role that these emerging technologies could play in the survey process through supervised autonomy. IACS UR Z17, titled "Procedural Requirements for Service Suppliers", creates a regulatory regime that permits the usage of RAS within the ambit of classification survey and is a strong reminder of the IACS's progressive and forward-thinking nature in this technological savvy era. Among the many important provisions, "control of data" as found in s. 5.2.6 of IACS UR Z17 is pertinent to the current discussion on data governance [62]. S. 5.2.6 provides that:

> "When computers are used for the acquisition, processing, recording, reporting, storage, measurement assessment and monitoring of data, the ability of computer software to satisfy the intended application shall be documented and confirmed by the service supplier. This shall be undertaken prior to initial use and reconfirmed as necessary." [62]

Demonstrating adequate control over data as noted in the above section is a precondition to the authorization and subsequent certification of service suppliers by the

concerned classification society. Under IACS UR Z17, RAS are permissible pursuant to Sections 3 and 16 of Annex I [62]. The former section applies to firms or service providers that have the capacity to carry out in-water survey on ships (and mobile offshore units) with the help of remotely operated vehicle (ROVs). The latter section applies to firms or service suppliers using RIT (ranging from UAV, ROVs, climbers, unmanned robot arm, drones, etc.) as an alternative to close-up surveys. A close-up survey is an examination "where the details of structural components are within the close visual inspection range of the surveyor" and is an integral formality satisfied, if required, during annual, intermediate and special (or renewal) surveys for detecting fractures, buckling, substantial corrosion, and other types of structural deterioration [63].

A further investigation into the two relevant Annexes of IACS UR Z17 reveals that a number of actors are associated with what could be termed as the data governance and management regime. The stakeholders called upon to remain engaged at various stages include the concerned classification society, flag state administration, as well as service supplier including supervisor, operator, and surveyor. While the classification society and flag state administration deals with approval and certification aspects, the core actors of the service robotic data regime involve the service supplier, who is under obligation to have documented data collection and storage procedures and guidelines in place with the verification of data acquisition tasks, which remain at the discretion of the attending surveyor. All in all, the supplier bears the main responsibilities of ensuring that platforms with data capture/collection/recording devices are readily available and of observing documented procedures that stipulate requirements for location attribution, validation, and storage of data [62].

### 3.5. Comparative Insight: RAS Data Governance Best Practices

The authors conducted a cross comparative examination with a view of reviewing data governance procedural requirements developed by selected individual member societies (see Appendix A, "Selection Criteria: Class Society Primary Sources on Drones, Crawlers and ROV"). Respondents (including service suppliers, selected classification societies, RAS experts) were also interviewed to confirm that the techno-regulatory provisions were interpreted appropriately. It is noteworthy that in the conduct of the proposed analysis, the IACS UR Z17 data requirement (examined previously) serves as the international evaluative standard against which individual member society rules were benchmarked. Existing rules in the form of best practices aimed at service providers are enshrined by nine individual member societies, i.e., Lloyds Register (LR), Bureau Veritas (BV), Det Norske Veritas (DNV), Registro Italiano Navale (RINA), American Bureau of Shipping (ABS), Russian Maritime Register of Shipping (RS), China Classification Society (CCS), Korean Register (KR), Nippon Kaiji Kyokai (NK); these were the principal materials that were examined out of the twelve IACS member societies (see Appendix A, "Selection Criteria: Class Society Primary Sources on Drones, Crawlers and ROV"). The rationale behind the above selection was guided by the objective of satisfying adequate regional coverage, with LR, BV, DNV, and RINA covering the European Union landscape; ABS representing the Americas; RS covering the Russian region in their own rights; and CCS, KR and NK covering the Asian region. All collected materials were incisively examined through the lens of unique developments.

The results of the cross-comparative examination revealed that while member societies acknowledge IACS UR Z17 as the foundation of respective rules developed in silos, not all selected member societies have explicit provisions on data, let alone innovative out-of-the-box rules. On a positive notion, the term remote inspection technology (RIT) is in principal usage in all classification society guidance documents. We also confirmed that the RIT referred to within guidance documents is synonymous with the term RAS.

In the quest for extracting unique developments, it is apparent that the document titled "Approval of Service Supplier Scheme" captures DNV's diligent effort in regulating the subject matter [64]. Developed in 2016, the above document covers "data storage"

rules in s. 16.1.4 of UR Z17's Appendix A. The section covers data storage in the context of RIT-survey reporting and furthers the requirement that all data files should be named after the structure surveyed, and should be stored by the service supplier and readily available at request from DNV for a duration of 5 years [64]. It was also noted that the said provision directly covers the "data storage and infrastructure" decision domain of the data governance framework.

A different approach was observed in the requirements tabled by LR. In the document titled "Procedures for Approval of Service Suppliers", LR upholds the general requirement on "data control" in s. 1.3.7 as found in IACS UR Z17 [65]. The same observation applies to RINA, given that the document titled "Rules for the Certification of Service Suppliers" follows IACS UR Z17 verbatim. LR has nevertheless developed separate guidelines for RIT as well as for unmanned aircraft systems. In terms of RIT, the document titled "Remote Inspection Technique Systems (RITS) Assessment Standard for use on LR Class Surveys of Steel Structure" covers data calibration and analytics, whereas the document titled "Guidance Notes for Inspection using Unmanned Aircraft Systems" provides niche guidance on drone operational as well as data capture and treatment considerations [66,67]. Although the latter document only covers a single RIT, i.e., drones, noteworthy provisions on data can be found in s. 8 entitled "Inspection Data" [67]. Axiomatic from the title, s. 8 branches out into three distinct data-relevant recommendations, with the most important one being s. 8.3, which highlights considerations on data security—an explicit decision domain category on the overarching data governance framework, as discussed earlier [67]. S. 8.3 stresses having appropriate "data security principles, standards and methods" with a view of ensuring that all data acquired receive security and protection against "manipulation or unwanted distribution" [67]. Within the texts of s. 8.3, express reference to ISO/IEC 15408 (on Information Technology–Security Techniques–Evaluation Criteria for IT Security) and ISO/IEC 27001: 2013 (on Information Technology–Security Techniques–Information Security Management Systems) has been provided.

On the American front, best practices are offered in the texts of "Guidance Notes on the Use of Remote Inspection Technologies" developed by the ABS in 2019 [17]. As stated in its "Foreword" note, the document is a holistic approach to governing UAVs, ROVs and robotic crawlers, taking into account rules and requirements as found in IACS Recommendations 42 and 76 and IACS UR Z17 [17]. A remarkable feature of the above document is the manner in which it governs remote inspection vehicle (RIV) post-operation data review and processing tasks, where RIV here is synonymous with both RIT and RAS. Sections 4.9 and 4.11 import detailed requirements that are absent in IACS UR Z17, and prescribe the to-dos in the stages of pre-inspection, during inspection, and post-inspection, touching upon all elements from the data decision domain directly or indirectly. It is important to note that while there is observed adequate emphasis on "data security policies and procedures" in Section 4.11.1.1(h), the paragraph notes that those policies and procedures are to be developed and maintained by the concerned service provider [17]. Whether or not ISO/IEC 15408 and ISO/IEC 27001: 2013 would have been appropriate references is a matter that requires further consideration by concerned members of the ABS.

Among the Asian counterparts, CCS utilizes two distinct documents: one on remote surveys and the other one on the usage of UAVs [68,69]. Of the two, the latter contains provisions on data and information in Section 3, which covers data acquisition, data processing, and data security aspects in a fashion similar to the ABS Guidance Note [68]. Indeed, Section 3 touches on the three stages of inspection (as mentioned above) and prescribes adherence with "statutory or regulatory requirements, company regulations and contractual agreements requirements" [68]. As for NK, the "Rules for Approval of Manufacturers and Service Suppliers" issued in 2020 incorporates IACS UR Z17 rule to the letter in Section 1.4.1, and hence the focus remains confined to the "data control" aspect [70]. The same is observed in the case of KR when observing the texts of Section 5(2)(f) (Annexes 1–11) as found in the document titled "Guidance Relating to the Rules for the Classification of Steel Ships" [71].

Finally, a thorough investigation into the Annexes of the "Guidelines on Technical Supervision of Ships in Service" published by the RS provided interesting insight into the domain of inspection using RIT, inter alia [72]. Unfortunately, neither the general provisions nor the main Annexes of the 2019 Guidelines prescribes data considerations for ship owners and service suppliers, other than prioritizing the basic conditions and procedures for the conduct of such operations.

*3.6. Hovering over RAS Governance Gray Areas*

The importance of good data governance is internationally recognized. Regions from all across the world have endeavored to regulate data through the adoption of protection and privacy legislation. According to the United Nations Conference on Trade and Development (UNCTAD), a total of 204 countries currently have either national "legislation" or national "draft legislation" (184 countries enacted legislation; 20 countries tabled draft legislation) in place [73]. In other words, statistics show that 66% countries developed legislation, 10% countries developed draft legislation, 19% countries have no legislation, and 5% countries are yet to provide information on the subject matter [73]. A cursory look at the titles of existing national legislation indicated that all status quo efforts remain within the parameters of personal data or personal information protection.

National emphasis to protect personal data is justified. Regardless of the ensuing tension created by the requirements, the big picture renders the international influence or "global reach" of GDPR as the substantiated ground on which this justification rests [74]. As for holistic efforts to safeguard data covering both personal and non-personal categories, six countries (Russia, USA, Australia, Norway, China, and the Netherlands) to date have managed to device cyber strategies [75]. With this in view, the question then is, what progress have the international actors made in the maritime sector?

In an effort to answer the preceding question, it becomes clear that emphasis on both personal and non-personal data security is omnipresent at the international level. Markedly, in the context of non-personal data, "cyber security" is the current buzzword in both the maritime and ocean domain [76]. Cyber security breaches through electronic virus and malware have disrupted commerce on more than one occasion, exposing the vulnerable face of the shipping industry [76]. Even the IMO mainframe has not been spared as it has already encountered a cyber-attack in 2020.

In retrospect, technology applications such as the automatic identification system (AIS), the global positioning system (GPS), the satellite-based long-range identification and tracking system (LRITS), as well as the vessel monitoring system (VMS), while collectively supporting the notion of safety and environmental protection, have also paved the way for discussions on illegal electronic interferences and disruptions, in turn prompting the feasibility of the above systems to be questioned. The inadequacy of the current international regime has called for the need to undertake further action by IMO Member States (MS) that adopted IMO Resolution MSC.428(98) titled "Maritime Cyber Risk Management in Safety Management Systems" in 2017. Bearing in mind the recommendations on maritime cyber risk management embedded within the IMO's "Guidelines on Maritime Cyber Risk Management", the 2017 Guidelines require the incorporation of safety management systems in a company's Document of Compliance by 1 January 2021 (International Maritime Organization, 2017). The 2017 Guidelines are apparently aligned with the overall objectives of International Ship and Port Facility Security (ISPS) Code, developed at the IMO under the International Safety of Life at Sea Convention as well as the International Management Code for the Safe Operation of Ships and for Pollution Prevention (ISM Code). Awareness remains at the heart of all international initiatives. However, we assert that awareness while serving as the stepping stone towards good data governance nevertheless requires in-tandem acknowledgment of a number of emerging challenges. The aforementioned statement thus begs the question: what are the existing challenges?

Self-evident from the above discussion, ongoing efforts are apparently concentrated on developing digital awareness among organizations and companies that are vessel-

centric. The IMO's current focus remains on maritime autonomous surface ships (MASS), which leaves the bulk of work on RIT regulatory standard development tasks at the hands of the IACS (Johansson, 2021). Although the IACS is closely tied to the IMO through a Memorandum of Agreement (MoA), there are currently no hints that international cooperation on the subject matter will be underway any time soon.

It is axiomatic that the current maritime service robotics data requirements are driven by IACS UR Z17. Prima facie, s. 5.2.6 is terse, rendering the IACS standards insufficient, unsettled, and incomplete. Data security and data storage considerations are overlooked, which are likely to segue into more complex challenges in the future once service robotics are in mass deployment after they are met with acceptance by all flag states and classification societies. This bleak situation is further exacerbated by the lack of a horizontal initiative by member societies to flesh out a much-needed data governance framework. In addition, the existing initiatives are disharmonized, given that there are no restraints that inhibit individual member societies integral to the IACS's "big 12" or the societies outside of the "big 12" from crafting their own class rules [60]. As was observed in the comparative insight segment (previously discussed), this practice is already constant. Some individual member states have promulgated standards that touch upon all the major issues pertinent to the data governance framework, and individual progress is commendable. The nature and scope of those standards are indeed commendable, but it is difficult to single out a document from any member society as containing the data governance "gold standards" for maritime RAS. Again, the member societies that have provided guidance on the subject matter do not cover all types of RITs. This in turn raises the question, do individual types of RIT deployed in vessel inspection and survey require separate attention?

The answer to the question posed above requires an assessment of two different strands of analysis: firstly, the difference in data generated by individual RITs, and secondly and distinctly, the sufficiency of a single ambient data governance framework that ties all important decision domain threads.

With a view of assessing the first strand, the end-product of RITs requires a separate focus. What is observed is that the current permissible limit of RIT deployment is limited to a close-up survey (and thickness measurement where required) of ships' structures and in-water survey (in lieu of docking survey) of ships' underwater structures. Despite the various types of RITs that are available to complete the tasks that remain within the above regulatory limitation, the principal objective is to acquire data for verification and validation by the principal surveyor. RITs have built-in image sensors that transform photons into electrical signals that are then viewed on high-definition display screens, recorded and analyzed by operators engaged in commercial inspection activities. Despite the various types of RITs, the principal objective of all technological deployment remains unchanged. Data, whether generated from a drone, magnetic crawler, or an ROV, still need to adhere to the IACS UR Z17 prescribed form, i.e., high-quality video images and still images, and therefore the format of the end-product will not vary.

Insight into the second strand unveils that a common data governance framework is sufficient to cover all types of RIT relevant actions. Examples of this are seen in the ABS's 2019 Guidance document, which offers a functional and consolidated approach. A close look at its s. 4.9 and s. 4.11 confirms that all "data review" and "data post-processing" provisions are one-size-fits-all and apply to the three ABS-approved remote inspection vehicles (RIVs), i.e., drones, crawlers, and ROVs. Whether or not new technologies with hybrid applications emerge in the future and require separate data governance framework is still a "wait and see" game.

In terms of data preservation, a topic that also needs attention is the duration aspect. As is tradition, once the data acquisition tasks are complete, the operator is under obligation to provide those videos and still images and data in a format acceptable to the attending surveyor for review and examination, after which survey reports are developed and submitted to the classification society for approval and certification. The operational procedures are shepherded by a contractual agreement entered into between ship owners

and service providers or ship owners and a classification society. What is apparent is that the IACS, as well as a majority of the individual classification societies that belong to the "big 12", are rather silent on post data acquisition steps, which might indirectly affect the data storage and security dimension. The current practice gives copyright ownership to the service providers, which is also coupled with the right to retain data for a limited duration within which data need to be communicated subject to request from the concerned classification society.

In reality, RITs are programmed to acquire vessel structural information that forms a part of the vessel-history. The vessel itself is a business asset, and from that standpoint, adequate protection should be given to safeguard the information so gathered in the operational process. While charter parties are only interested in assurance of vessel seaworthiness, should the data containing structural defects fall into the wrong hands, unforeseen negative effects could ensue. Shipping is a competitive industry, and that is why asset-related information should be treated with utmost confidentiality. Although not a contentious issue at this point, nonetheless a number of topics require further clarification: Who should retain the copyright ownership of data gathered from RITs? What is the duration of preservation of data and image from close-up and in-water surveys? Should there be any safeguard mechanisms for service providers against third-party liability? These are a few outstanding matters that require consideration in order to create a level playing field for all stakeholders involved in the RIT business model.

## 4. Discussion Based on Priori Synthesis: Ways Forward

The findings from BUGWRIGHT2 identified a set of twenty-six action items that comprise the international regulatory blueprint (see Table 2). The blueprint accommodates cardinal elements while bearing in mind the mandate of relevant international actors with law-making, technical, and operational capabilities when maneuvering in the so-called business model landscape. All elements address the inconsistencies and barriers that have the potential to impede the successful deployment of emerging service robotic technologies for taking evidence-based decisions in the process of vessel inspection, survey, and maintenance. In exploring niche areas, the propositions tabled in eight out of the twenty-seven items of the blueprint is predicated on the view that data are the nucleus around which the business-model rotates. Therefore, the role of the all actors involved in data acquisition, data processing and analysis, and data validation requires acknowledgment followed by appropriate review and revision.

**Table 2.** Overview of the regulatory blueprint action items for harmonization of international arrangements (synthesis derived from BUGWRIGHT2 Deliverable 1.4.1).

| Underlying Element | Action Item |
|---|---|
| Interorganizational consultation | **Action Item 1 in conjunction with Action Item 2:** Creation of a forum to take part in revisions and reforms undertaken by IACS in relation to Unified Requirements. |
| Categorization based on capacity | **Action Item 3:** Classify based on capacity and determine whether MAVs, AUVs, and crawlers fall under the scope of "mobile robots". |
| Standalone definitions | **Action Item 4:** Consider standalone definitions (in the following manner) for MAVs, AUVs, and crawlers rather than referring to all technologies under the overarching term "remote inspection technologies". |
| Classification pursuant to degree of autonomy | **Action Item 5:** In order for procedural rules and requirements to keep pace with technological innovation (towards full autonomy), service robots require a form of categorization along the lines of degree of autonomy. A potential way forward is to follow closely the degrees rendered to vessels and how the different stages were set by IMO's MASS (as the first step in scoping exercise). |
| Operational limitations and conditions for service robotics | **Action Item 6:** Consider operational limitations for MAVs, AUVs, and crawlers that help ensure effective completion of survey process. <br> **Action Item 7:** Consider pre-operation, in-operation, and post-operation conditions for service robots. |

**Table 2.** *Cont.*

| Underlying Element | Action Item |
|---|---|
| Expand existing provisions on "alterations" | **Action Item 8:** Revise provisions in relation to survey and inspection planning, and consider all potential risk assessment options. <br> **Action Item 9:** Indicate in detail the procedures in cases where the service supplier alters the certified system, which in turn affects the quality system. <br> **Action Item 10:** Consider revision to include provisions on certification of multisite organizations under the rules concerning certification. <br> **Action item 11:** Consider revision of existing provisions related to survey procedures) to include a client service system (CSM), live-streaming during remote survey, and real-time collection of survey process information, solutions in case of problem with live-streaming, recording of process and conclusion in the ship log, and conditions for certification. <br> **Action item 12:** Consider technological platforms for facilitation collection and delivery of survey-related information such as computers, intelligent remote glasses, and digital cameras for live-streaming purposes. <br> **Action Item 13**: Consider using more a factual and objective post-reporting system. |
| Safety management system | **Action Item 14:** Consider incorporating valuable provisions related to a safety management system, with explicit reference to safety policy, safety risk management, safety assurance, and safety promotion. |
| Liability clause | **Action Item 15:** Consider incorporating a liability clause in UR Z17 for maintaining third-party liability insurance in case of accidents or deaths. |
| Data governance and management | **Action Item 16:** Hull inspection data should be kept confidential as this may constitute a trade secret for the shipowner. Legitimate practices should be in place for the collection, storage, and use of unpublished data of economic significance. For overcoming barriers to data governance and data management, explicit provisions are needed in the form of a contract that specifies the allocation of responsibilities and roles for the ownership, storage, security, and sharing of information between service suppliers, classification societies, and shipowners. Sound data governance principles are essential to help minimize risks and keep external cyber securities threats out of their networks. The contract should be signed during the planning stage of hull inspection. <br> **Action Item 17:** IACS guidelines should elaborate on their provisions about data management from the use of remote techniques. <br> **Action Item 18:** Data ownership, which is one of the most critical parts of the data governance process, defines the rightful owner of the data elements, sharing policy, and access rights to third parties granted by the data owner. During the planning stage of hull inspection, a clear understanding between service suppliers, classification societies, and ship owners/managers should be maintained about data ownership. <br> **Action Item 19:** Digital data preservation gives reliable protection to information and systems needed to ensure the long-term usability of data and metadata. Clear allocation of responsibility should be given to the party that is responsible for data and image preservation. <br> **Action Item 20:** Distributed data between the different stakeholders intensifies data security efforts between participants in the data process. Cloud environments encounter increased security threats due to inadequate access management and system vulnerabilities. Measures should be in place for the security and confidentiality of remote inspection technology data by all the relevant stakeholders to ensure a sound data governance process. <br> **Action Item 21:** During the planning stage of hull inspection, it should be specified how the data are shared between the different stakeholders to ensure secure data transfer between data owners and users. Provisions should exist about the sharing of data in the formal agreement. A secure industry platform could be utilized for secure data transfer between data owners and users, when saving and sharing the video stream from the remote survey. <br> **Action Item 22:** The current copyright regime does not protect computer-generated data; thus, explicit provisions in the contract should be made to safeguard computer-generated works. <br> **Action Item 23:** Potential liability issues that stem from the use of data should be underlined in the contract. Input material supplied by the asset owner to the service supplier before the hull inspection (i.e., images, drawings, and designs) should not infringe the copyright or other rights of a third party. |

**Table 2.** *Cont.*

| Underlying Element | Action Item |
|---|---|
| Harmonizing statutory and class rules with reference to close-up survey | **Action Item 24:** Consider aligning the definition of a "close-up survey" found in IMO's Enhanced Survey Programme, given that the current definition of close-up survey is inadequate as the IACS created the possibility to use RITs for remote inspection, allowing the surveyor to conduct close-up surveys through sensors. |
| Controlling variety for optimum quality performance by regulating technical and operational standards | **Action Item 25:** Develop a methodology to establish standards based on product categorization with the aim of reducing variety to identify the best product from all categories of RITs. |
| Creating a remote inspection technology "trustworthy ecosystem" | **Action Item 26:** There should not only be efforts to market quality products so that smooth integration is possible but also efforts to create lawful, ethical, and robust service robots that can render end-users trust in the products deployed for survey and maintenance tasks. |

The first data-themed action item furthered in the regulatory blueprint is a recommendation calling for the security and protection of all data acquired in relation to the structural elements of a vessel. The recommendation tied to the first action item anticipates dialogue and discussion among ship owners and all other actors noted within IACS rules and requirements. Based on constructive exchanges, it should be confirmed whether legitimate practices should be in place for the collection, storage, and use of unpublished data of economic significance. The platform should support recommendations to integrate sound data governance principles in the context of the data decision domains to safeguard information from cyber security threats—a common phenomenon that could lead to unforeseen damages.

The second and third thematic items are founded on a simple notion: IACS is the principal actor that governs sound data governance system for service robotics. However, a consideration of the requirements implemented by other international organizations is important to address incidental loopholes. LR, for example, has referenced ISO/IEC 15408 (Information Technology–Security Techniques–Valuation Criteria for IT Security) and ISO/IEC 27001:2013 (Information Technology–Security Techniques–Information Security Management Systems) in the procedural document that highlight valuable tools for stakeholders in the development of secure IT systems. This invokes the thought whether IACS, as the lead international organization on the topic, should adhere to the same. Through the second item, the authors assert that harmonization can be achieved through an explicit reference of ISO in both IACS Guidelines No. 42 and IACS UR Z17, especially considering the fact that a majority of the countries are concerned with GDPR and personal data, which leaves non-personal data unprotected and vulnerable.

The third item then focuses on finding a pathway to determine data ownership—an important "detail" that should be considered and integrated into the general provisions of IACS UR Z17 procedural requirements. It is through this third action item recommendation where we submit that there should be a clear understanding of the concept "ownership" among service suppliers, classification societies, and ship owners/managers. Generally speaking, "data ownership" is set by the enterprise's upper echelons and is related to the allocation of responsibilities over data [77]. The ownership decisions encompass a wide range of issues such as domains, data availability, accessibility, and frequency of updates [78]. The outcome of the third action item, according to our views, should help determine whether the fourth action item (i.e., incorporation of a separate section on data security and protection) is required within Section 3 and Section 16 of Annex I of IACS URZ17. We hold the position that clear provisions on data control and data security should exist; otherwise, the data governance framework may suffer. Determining which stakeholder organization should retain data ownership when working within a dynamic business model certainly has positive implications. A quality system under ISO 900 series, referred to in documents developed by individual member societies, prescribes accountability as a part of the engagement process under the quality management principles. Provisions on data control and data protection would certainly help monitor data-centric activities by

holding the organization responsible for data control and protection accountable, which is a sound way forward.

Incorporating the much-needed provisions in the texts of IACS UR Z17 would also create the right set of circumstances to review the current contractual practices (see Figure 3). This comes as a timely recommendation and serves as the fifth action item. The lack of adequate coverage of the specifics has paved the opportunity for service suppliers to develop contracts based on convenience and as seen fit. The roles and responsibilities for data ownership, storage, security, and sharing of information remain not catered for to say the least and require also an in-depth review of all private contracts developed by service suppliers. It is important to enquire how data are preserved and whether the service suppliers have adequate and reliable mechanisms in place to ensure the long-term usability of data and metadata gathered from RITs and ROVs.

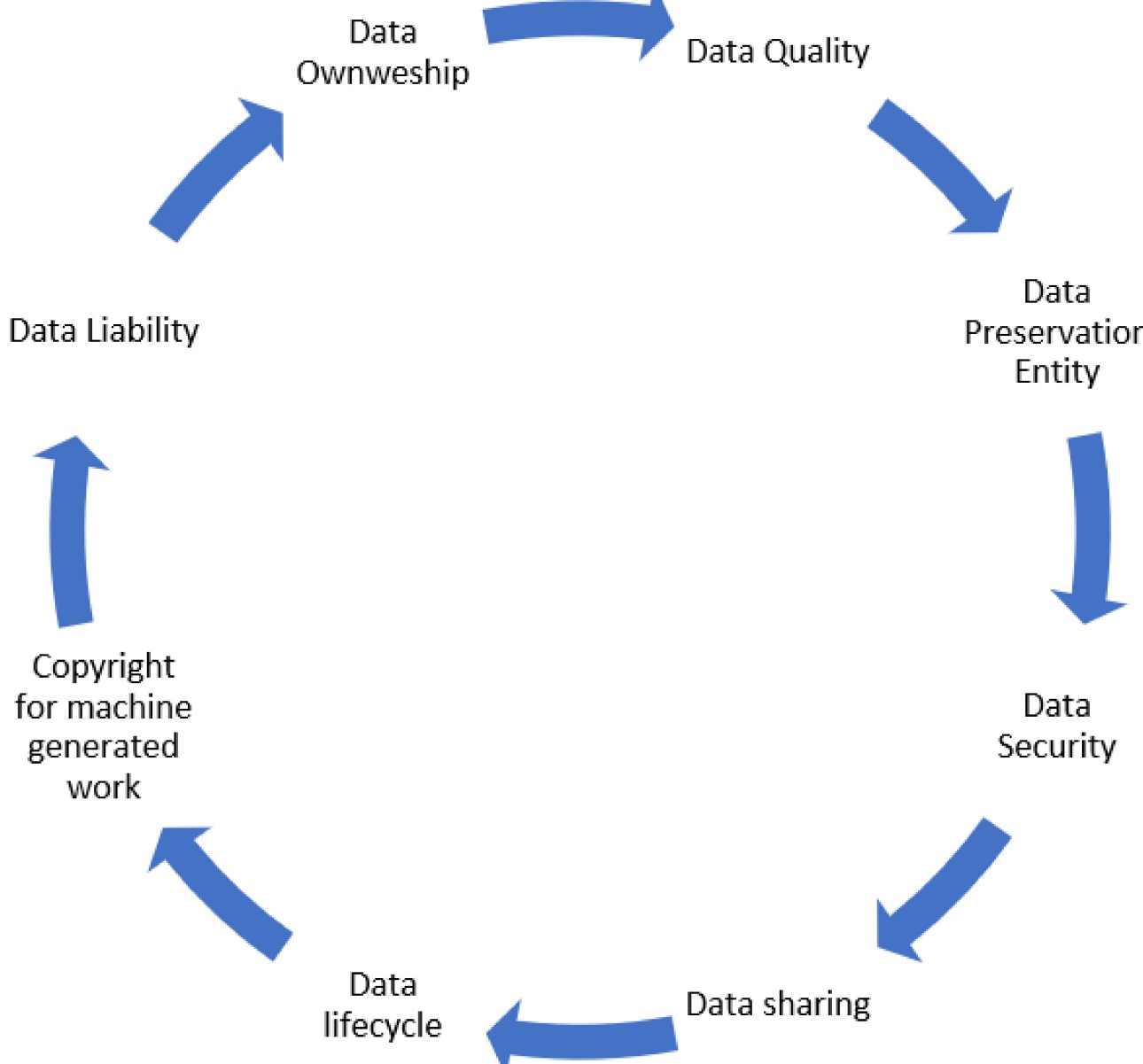

**Figure 3.** Visual-cycle of the data elements to be included in the contract between service suppliers, classification societies, and asset owners/operators.

Connected to the fifth action item is the need to verify whether organizations in the business model, especially service suppliers have a well-documented organizational "information systems security policy" and "backup strategy", both of which comprise the sixth action item, followed by feasible recommendations for consideration. For the former, a viable way forward could be that the entities of the business model implement tools and techniques to prohibit unauthorized access to programs and information resources. Users should have a unique user identifier (through passwords or other authentication mechanisms) to keep track of users for establishing accountability. As for the latter, i.e., a backup strategy, organizations could consider a digital infrastructure containing schedules for routine backup check and fast recovery tools. Again, one should come across ISO/IEC 15408 and ISO/IEC 27001:2013, which articulates general principles that could inspire a panoply of thoughts leading to concrete solutions post-verification and in cases where both security policy and backup strategy is absent.

The seventh listed action item in the regulatory blueprint corresponds to data transfer and secured sharing. Prior to the commencement of the operational phase, the IACS prescribes a planning stage with a number of items that are discussed at length by the stakeholders involved in the business model. We believe that specific discussions should be aimed at the best ways to share data among the stakeholders with a view to ensuring secure data transfer between the "owner of the data" and the others including end-users. Whether or not this item could encapsulate the incorporation of provisions alongside data control and data protection, or whether this should be covered under the formal contract between service suppliers and ship owners remain at the discretion of the international actors. Notwithstanding, we opine that this is an important step because this also concerns data security, and therefore a secured platform for data transfer between data owners and users could be the best way forward when saving and sharing the video stream from the remote survey. In doing so, the party providing access should implement and administer access restrictions to ensure that only authorized individuals have the ability to access or use information resources. The use of the universal serial bus (USB) for data sharing, according to the authors, should be avoided.

The final action item of the data segment relates to liability and contains a precautionary approach in order to avoid legal consequences. Caution should be exercised by the asset owners when sharing "input material" with the service supplier before the survey, e.g., images, drawings, and designs, lest this infringes the copyright or other rights of a third party. In case of infringement, the service supplier should be held unaccountable against any loss, damage, or other claims arising from such violation. On the other hand, service suppliers should be discouraged from using survey data for marketing reasons without the prior approval of the asset owner. It is understood that the specifics are currently absent in IACS UR Z17, paving the opportunity for service suppliers to develop contracts based on convenience and as seen fit. The roles and responsibilities for data ownership, storage, security, and sharing of information remains ambiguous to say the least. A possible way forward, should IACS deem it appropriate, is to mark out ways to strengthen due diligence on the part of service providers when holding the copyright or ownership of data obtained through the usage of RITs or ROVs.

## 5. Conclusions

Approximately 100,000 commercial vessels with a displacement of more than 100 tons navigate the oceans and the seas worldwide [79]. Altogether, there are more than 9000 large ships and more than 4000 very large ships that are above the age of five years [80]. Ship owners need to ensure that these vessels are fit for service, which comes off as a requirement under the UNCLOS [80]. Obtaining certification for all class and statutory surveys provides evidence that all survey and inspection related procedures are complete. However, the procedures to ensure fitness depend on the type of survey (annual surveys need around 1–2 days, intermediate surveys take around 3–4 days, specials surveys go around 1–2 weeks), and there is a need to also factor in post-inspection actions to mitigate defects

and deterioration. Maritime RAS could just well be the time savior in a world where shipping is conducted in a just-in-time fashion.

Regardless of the many opportunities that surface with maritime RAS technological applications, a well-implemented check and balance system should continue to operate in tandem with classification and statutory tasks. The topic of maritime non-personal data needs separate attention, broadly owing to the fact that a majority of the world's national strategies are struggling to protect data. Alongside personal data, non-personal data protection issues have already emerged, prompting the adequacy of the current data governance framework to be questioned. Questions are directed at international organizations that have an obligation to ensure that legal provisions aimed at governing technological products are free from imbalance and asymmetry. The success of the science–policy interface for maritime non-personal data requires a review of all the building blocks to ensure that asset related information is protected throughout the entire lifecycle. Generic problems and questions on non-personal data protection are likely to recur in other instances once fully autonomous ships using e-navigation and other forms of autonomous communication to establish contact between and among vessels are in place. This could invoke bring in other questions, such as the feasibility and effectiveness of current big data analytical techniques for an increased rate and volume of data—an issue of significant importance in offshore environments that needs to be considered as the industry moves towards the Internet of Shipping [80–85].

It is also necessary to clearly highlight that there are more desired capabilities in the pipeline, such as autonomous maritime RAS that connect automatically with autonomous vessels through machine-to-machine communication and are guided by the principles of AI. As scientists, engineers, and technologists follow the AI route and usher in the advent of the sixth-generation wireless (6G) communication (i.e., the successor of the fifth-generation wireless) for discharging "unprecedented capacity and latency", there needs to be a solid protection regime to avoid unwanted exposure of data and liability issues [81]. It can be said that 6G communication envisages a tech-savvy environment that has the potential to blur the understanding between human-to-machine and machine-to-machine interactions. We speculate that 6G itself combined with the groundbreaking concept of Internet of Everything (IoE) is likely to be labyrinthine in nature and is expected to contain hidden layers of communication that may require countermeasures to protect the data decision framework as well as the necessary building blocks for smooth user interface. However, until the introduction of 6G and IoE, engineers and policy makers need to establish a settled discourse of data integrity in maritime and ocean domain by observing the current IoRT architecture.

In reality, a sound data governance approach goes beyond just the protection factor simply because the data decision dimension and building blocks secure a "trustworthy" environment. A trustworthy business model entails trust in the product that is likely to have a positive repercussion among the stakeholders that are a part of the business model of non-personal data [82,83]. The current system is characterized as supervised autonomy (see Table 1). Detecting inherent surreptitious vices is a pre-condition in order to help engineers and manufactures of maritime RAS pass the design bottleneck and to ensure market success and subsequent successful integration by flag states and classification societies into mandatory survey processes. The key action as envisaged by authors is to flag out data-related liability provisions in international standards, and when doing so, consider the theoretical constructs of a sound data governance framework before harmonizing the international RAS data management system with other available best practices. Should the action materialize, the benefits are likely to be reaped by the actors involved in the business model, thus creating an environment where each and all are fully aware of the responsibilities and liabilities, whether strict, absolute, or vicarious

Here, collaboration among different international stakeholders of the maritime RAS ecosystem is climacteric. This could include discussions among all international bodies, including organizations that have RAS and maritime RAS mandate, e.g., IMO, IACS,

ISO, engineers, policy makers, flag state authorities, manufacturers service suppliers, classification societies, and ship owners and operators to flesh-out solutions on this topic. All in all, what must be hindered is the development of maritime RAS technology-based laws, rules, and requirements in silo. Harmonization has always been at the epicenter of international efforts. Therefore, cultivating international harmonization through optimal solutions that answers the current burning questions leading to a regime that safeguards non-personal data is in order. After all, it is not only that the world contemplates a future where there exists maritime RAS-to-autonomous vessel communication, but also there is a silent aspiration that works to support safe and liability-free maritime RAS-to-autonomous vessel communication. Still much work lies ahead to keep the implications positive in mankind's epic era of combating to establish digital serenity and its unfortunately already inevitable impacts.

**Author Contributions:** T.M.J., as the WMU Principal Investigator of BUGWRIGHT2 was responsible for developing the main draft, which was based on raw data provided by A.P. Significant revisions were made by D.D. in the development of this article. All authors have read and agreed to the published version of the manuscript.

**Funding:** This paper is derived from research conducted under the European Union (EU) Horizon 2020 funded project titled Autonomous Robotic Inspection and Maintenance on Ship Hulls (BUG-WRIGHT2) under grant agreement No. 871260.

**Institutional Review Board Statement:** Not applicable.

**Informed Consent Statement:** Not applicable.

**Data Availability Statement:** Not applicable.

**Acknowledgments:** The authors would like to thank Ronán Long and Clive Schofield of the World Maritime University–Sasakawa Global Ocean Institute and the Nippon Foundation for their generous support. The authors are grateful to Anastasios Tsalavoutas and David Knukkel for providing first-hand information incorporated in the section titled "RAS and the Internet of Robotic Things: Through the Prism of Data".

**Conflicts of Interest:** The authors declare no conflict of interest.

## Appendix A. Tabular Amalgamation of Published Literature and Primary Sources Based on Selection Criteria

| Selection Criteria: Literature on Data and Information | |
|---|---|
| **Author(s)** | **Title (Year of Publication)** |
| Zins, C. | What is the meaning of "data", "information", and "knowledge"? |
| **Selection Criteria: Literature on Data and Asset** | |
| **Author(s)** | **Title (Year of Publication)** |
| Lake, P.; Crowther, P. | Data an Organizational Asset. In *Concise Guide to Databases* (2013) |
| **Selection Criteria: Literature on Big Data** | |
| **Author(s)** | **Title (Year of Publication)** |
| Boyd, D.; Crawford, K. | Critical Questions for Big Data: Critical Questions for Big Data (2012) |
| Zhang, S.; Gao, H.; Yang, L. and Song, J. | Research on big data governance based on actor-network theory and Petri nets. In IEEE 21st International Conference on Computer Supported Cooperative Work in Design (CSCWD) (2017) |
| Johansson, T.; Long, R.; Dalaklis, D. | The Role of WMU-Sasakawa Global Ocean Institute in the Era of Big Data (2019) |

| Selection Criteria: Data Governance | |
|---|---|
| **Author(s)** | **Title (Year of Publication)** |
| McGilvray, D. | Data governance: The Missing Link for Data Quality Success. San Francisco Bay Area (2006) |
| Khatri, V.; Brown, C.V. | Designing data governance. *Communications of the ACM* (2010) |
| Al-Badi, A; Tarhini, A.; Khan, A.I. | Exploring Big Data Governance Frameworks (2018) |
| Al-Ruithe, M.; Benkhelifa, E.; Hameed, K. | Data Governance Taxonomy: Cloud versus Non-Cloud. *MDPI, Open Access Journal* (2018) |
| Al-Ruithe, M.; Benkhelifa, E.; Hameed, K. | A systematic literature review of data governance and cloud data governance. Personal and Ubiquitous Computing (2019) |
| Benfeldt, O.; Persson, J.S.; Madsen, S; | Data Governance as a Collective Action Problem. *Information Systems Frontier* (2020) |
| Janssen, M.; Brous, P.; Estevez, E., Barbosa, L. S and Janowski, T. | Data governance: Organizing data for trustworthy Artificial Intelligence (2020) |
| Chakravorty, R. | Common challenges of data governance (2020) |
| Thuraisingham, B. | Cloud Governance. In *IEEE 13th International Conference on Cloud Computing (CLOUD)* (2020) |
| **Selection Criteria: Data Privacy and User Rights** | |
| **Author(s)** | **Title (Year of Publication)** |
| Mattoo A.; Meltzer, J. P. | International data flows and privacy: the conflict and its resolution. *Journal of International Economic Law* (2018) |
| Chatzimichali, A.; Chrysostomou, D. | Human-data interaction and user rights at the personal robot era. In 4th International Conference on Robot Ethics and Standards: ICRES 2019, (2019) |
| **Selection Criteria: Personal and Non-Personal Data** | |
| **Author(s)** | **Title (Year of Publication)** |
| Finck, M.; Pallas F. | They who must not be identified-distinguishing personal from non-personal data under the GDPR (2020) |
| Somaini, L. | Regulating the Dynamic Concept of Non-Personal Data in the EU: From Ownership to Portability (2020) |
| **Selection Criteria: Actor–Network Theory** | |
| **Author(s)** | **Title (Year of Publication)** |
| Montenegro, L. M.; Bulgacov, S. | Reflections on actor-network theory, governance networks, and strategic outcomes (2004) |
| Latour, B. | *Reassembling the Social: An Introduction to Actor-Network Theory* (2005) |
| Dincer, D. | The Act-Shifts Between Humans and Nonhumans in Architecture: A Reading of Bruno Latour's Actor-Network Theory (2020) |
| **Selection Criteria: Internet of Robotic Things** | |
| **Author(s)** | **Title (Year of Publication)** |
| Ray, P.P. | Internet of Robotic Things: Concept, Technologies and Challenges (2017) |
| Yousif, R. | A Practical Approach of an Internet of Robotic Things Platform, Master of Science Thesis, KTH Royal Institute of Technology (2018) |
| **Selection Criteria: IACS Primary Sources** | |
| The International Association of Classification Societies | UR Z17, Procedural Requirements for Service Suppliers (1997) |
| The International Association of Classification Societies | Guidelines for Surveys, Assessment and Repair of Hull Structure–Corr. 1 (Recommendation 76) (2008) |

| Selection Criteria: Class Society Primary Sources on Drones, Crawlers, and ROV | |
|---|---|
| **Author(s)** | **Title (Year of Publication)** |
| Lloyds Register | Guidance Notes for Inspection Using Unmanned Aircraft Systems (2016) |
| Lloyds Register | Remote Inspection Technique Systems (RITS): Assessment Standard for Use on LR Class Surveys of Steel Structure (2018) |
| American Bureau of Shipping | Guidance Notes on the Use of Remote Inspection (2019) |
| Bureau Veritas | Approval of Service Suppliers (2020) |
| China Classification Society | Guidelines for Use of Unmanned Aerial Vehicles (2018) |
| China Classification Society | Rules for Classification of Sea-going Steel Ships (2015) |
| Det Norske Veritas | Approval of Service Supplier Scheme (2016) |
| Korean Register | Rules for the Classification of Steel Ships (2017) |
| Nippon Kaiji Kyokai | Rules for Approval of Manufacturers and Service Suppliers (2020) (Part 1 Chapter 1) |
| Registro Italiano Navale | Rules for the Certification of Service Suppliers (2020) |
| Russian Maritime Register of Shipping | RIT Requirements: Annex 39 (Guidelines for the Use of Remote Inspection Techniques for a Survey of Ships and Marine Structures) |
| Russian Maritime Register of Shipping | ROV Requirements: Annex I (Procedure for In-water Survey of Ships and Offshore Installations) of Guidelines on Technical Supervision of Ships in Service, Russian Maritime Register of Shipping |
| **Selection Criteria: Data Management** | |
| **Authors(s)** | **Title (Year of Publication)** |
| Loshin, D. | *Master data management* (2008) |
| Vilminko-Heikkinen, P.; Pekkola, S. | Changes in roles, responsibilities and ownership in organizing master data management (2019) |
| Earley, S.; Henderson, D.; Data Management Association | *DAMA-DMBOK: Data Management Body of Knowledge* (2017) |
| Brous P.; Janssen M.; Vilminko-Heikkinen R. | Coordinating Decision-Making in Data Management Activities: A Systematic Review of Data Governance Principles (2016) |

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
