# Peer review of "Maritime Robotics and Autonomous Systems Operations: Exploring Pathways for Overcoming International Techno-Regulatory Data Barriers"

_jmse, doi:10.3390/jmse9060594_

Round 1

Reviewer 1 Report

Very interesting topic with a great future perspective, which will create new and interesting contributions in the short and medium term. My personal opinion is that the "classes" are currently addressing the digitization of inspections and audits, they are at an early stage. Perhaps some reference to this evolution would be advisable. A reference to the GBS (Goal Based Standards), another line of work with great potential for the "classes", would also be positive. In any case great article!.

Author Response

General Comments: Very interesting topic with a great future perspective, which will create new and interesting contributions in the short and medium term.

We are thankful for the reviewer’s positive outlook in relation to the paper. Greatly appreciated.

Specific Comment 1: My personal opinion is that the "classes" are currently addressing the digitization of inspections and audits, they are at an early stage. Perhaps some reference to this evolution would be advisable. A reference to the GBS (Goal Based Standards), another line of work with great potential for the "classes", would also be positive. In any case great article!

We sincerely thank the reviewer for the comment. The authors have revised the paragraph in pp. 13, para. 2 with explicit reference to the evolution of class societies and GBS:

In this discussion the role of classification societies cannot be ignored. More than fifty individual classification society members have been quite influential actors within the shipping industry as far as the history of global maritime trade is concerned. Consequently, classification societies play a crucial role in meeting standards, with the focus recently shifting towards Goal-based Standards (GBS) through the development of standard rules and regulations in direct conformity with United Nations Convention on the Law of the Sea of 1982 (UNCLOS). It is not a coincidence that according to the official IMO website “Goal-based standards (GBS) are high-level standards and procedures that are to be met through regulations, rules and standards for ships.  GBS are comprised of at least one goal, functional requirement(s) associated with that goal, and verification of conformity that rules/regulations meet the functional requirements including goals.  In order to meet the goals and functional requirements, classification societies acting as recognized organizations (ROs) and/or national Administrations will develop rules and regulations accordingly. These detailed requirements become a part of a GBS framework when they have been verified, by independent auditors and/or appropriate IMO organs, as conforming to the GBS”.

The title under which classification societies operate in the process of developing GBS is known as Recognized Organizations (RO) – a title spread across a significant number of IMO conventions covering environmental and safety facets. Of the many international organizations that contribute to the creation of global maritime technology-based regulatory standards on classification and statutory rules, the IACS is viewed as the leading non-profit international body. Although the International Organization for Standardization has since 1947 played the traditional role of developing technological standards for robotics and robotic products, IACS’s parallel efforts to date is observed as being the lead in the creation of a techno-regulatory environment for the maritime industry. In the evolution continuum, it is noted that IACS has come a long way in so far as it’s promulgated standards cover more than 90% of the global cargo carrying tonnage …”.

With a view to brining more alignment, the authors have added the words “those emerging technologies” in p. 14, para. 3:

“Currently, the state-of-the-art survey and certification provisions reside among the various Unified Requirements (UR) developed by IACS, such as, UR Z3, UR Z7.1, UR Z7.2, UR Z10.1, and UR Z10.2. Notwithstanding the plethora of rules and requirements, IACS has duly considered the role those emerging technologies could play in the survey process through supervised autonomy”.

Reviewer 2 Report

This thesis conducted a comparative study by combining doctrinal methods and comparative methods. Journal of Marine Science and Engineering is a journal for research related to marine science and engineering. Unfortunately, this paper is suitable for social sciences, and no engineering elements were found in this paper. In addition, experimental and theoretical results cannot be confirmed. However, this study covers a sufficiently interesting topic, so browsing the right journals will yield good results.

Author Response

General Comments: This thesis conducted a comparative study by combining doctrinal methods and comparative methods. Journal of Marine Science and Engineering is a journal for research related to marine science and engineering. Unfortunately, this paper is suitable for social sciences, and no engineering elements were found in this paper. In addition, experimental and theoretical results cannot be confirmed. However, this study covers a sufficiently interesting topic, so browsing the right journals will yield good results

We thank the reviewer for the comment.

At the article-construction level, we developed an outline that discusses and explains developments on a topic, i.e., data, that originally belongs to the scientific discipline. The current framework of this paper is such that it has gone beyond the silo-based borders of science and became interdisciplinary with law in the form of international standards governing “data” – purely scientific in nature and originally generated through RIT equipment.

The authors believe that the special edition to which this article is submitted (Topic on “data” being one of the subject matters to this special edition as highlighted on the website) is the appropriate platform to disseminate the first-hand results of project BUGWRIGHT2 from which all stakeholders, including engineers and technical experts could benefit from. Section 3.3 covers the technical part of the article and one that acts as a strong foundation to all subsequent discussions. The original plan was to treat s. 3.3 as the solid basis with more technical details, but opted not to expand too much since that would create another paper within our paper and would rather compensate with first-hand self-explanatory photos from the source.  

We also believe maritime RAS is the future, and in that future, engineers need to be cognizant of incidental impediments that could result in hindering innovation and market growth. Observing the categories of audiences and readers of this Journal, then authors felt this article, first of its kind, requires the much-needed attention of those specialized audiences.

In terms of experimental results for a topic that revolves around techno-regulatory barriers, we will know more once RIT’s are in mass deployment. That being said, the barriers are apparent as “liability” resides in almost every discipline and has slowly enveloped the scientific community. It does not require the conduct of an explicit experiment, rather a cross-examination of those standards promulgated at the international level. Those standards as they stand today, have been amended several times, and assuming the introduction of RIT standards in the current texts – we can safely assert that they have appeared today as a result of much “consideration” – a word that borders observation based on experiments conducted by classification societies. Furthermore, the issues are being discussed and have been discussed in EU project platforms, confirmed by our colleagues that have been engaged in those projects. The authors are currently in the process of developing a “code” that comprises liability, trust and data governance, which will be taken into account in the relevant international platform (IMO) in 2022 (this information is shared in confidence as it provides justification on our part).

As for theoretical results, we have based s. 3.1 on those theories that are widely cited in academia. Data-liability is such a topic that defies theoretical results as there are foreseen and unforeseen consequences. However, the basic tenets remain the same and one of the pillars of the tenet is a “safety-net” for manufacturers and engineers that operate the RITs.

Finally, the authors would like to emphasize that the title has been refined, as we as well as several sections, including method and material so as to make it clear that we are dealing with a technological-regulatory topic that is a blend of the best of two worlds. As stated, (in the new addition) in p. 3 (lines 143-148): It is important that service robotics engineers, as they move from “human-in-the-loop” technology to “full autonomy” technology, remain aware that technology and law reside in the same continuum and therefore, interrelated. Technology should progress, but in that process, it must be ensured that such progress does not breach or violate any con-temporary legal provisions that are globally sensitive in nature – we strongly stand by our projection that the broader academic community will benefit from this work.

Last but not least, the authors are grateful that the reviewer finds “this study covers a sufficiently interesting topic”.

Reviewer 3 Report

Broad comments. The authors have made a concise overview of the topic focusing on. As a general drawback I could say that main research question has not been well established. The main objective of the work is not clear. Authors could try to clarify their motivation and the focus of the paper as it seems more as a list of legislation and definitions most of the time.

Specific comments. In general, the text is well structured and has clearly defined topics. The abstract is a good guide for what follows, even though could be refined such that it describes the motivation of the paper. More or less all fundamental theory details that are needed are discussed and concluding remarks are sufficient.

Some comments for improvement:

  1. Authors could consider use a more descriptive title for their work such that the exclusion of the “legal research” to be more obvious to the reader.
  2. In addition, authors could consider adding a description on the number of published papers and the way for the selection of the specific etc.
  3. Do authors have information regarding the technical requirements for data processing or time consumption of algorithms? Such information would be of great importance.
  4. Both Table 1 and Table 2 could be refined in order to better present relevant information.
  5. While authors try to explain the importance of data in maritime sector they do not mention the different difficulties due to the adverse conditions offshore. For sure this is one of the most important limitations towards the direction of Internet of Shipping (e.g. [1, [2]).

[1] Theodoropoulos, P.; Spandonidis, C.C.; Themelis, N.; Giordamlis, C.; Fassois, S. Evaluation of Different Deep-Learning Models for the Prediction of a Ship’s Propulsion Power. J. Mar. Sci. Eng. 2021, 9, 116. https://doi.org/10.3390/jmse9020116

[2] Spandonidis C.; Theodoropoulos P.; Giordamlis C. Combined multi-layered big data and responsible AI techniques for enhanced decision support in Shipping In Proceedings of The International Conference on Decision Aid Sciences and Applications 2020 (DASA20), (pp. 669-673). IEEE.

  1. In section 4 authors are mentioning 26 items that compromise the international regulatory blueprint. Eight of them are further discussed. A table or figure with relevant info for all items could be useful. In addition, authors could consider refining the legend of Figure 3 The source could be removed.
  2. Finally, since this effort lies in the borders of a review paper and a conceptual article, I would say that authors need to refine the paper, to point out the key message and the potential benefits of their work.

Author Response

Broad comment 1:

The authors have made a concise overview of the topic focusing on. As a general drawback I could say that main research question has not been well established.

The authors have now developed an explicit research question embedded in p. 3, para. 3:

“… grappling with two pragmatic questions: what are the thorny issues that could invoke data liability in an RAS-led vessel survey and inspection operation, and what are the pathways through which those issues could be mitigated?”

Broad comment 2:

The main objective of the work is not clear. Authors could try to clarify their motivation and the focus of the paper as it seems more as a list of legislation and definitions most of the time.

Focus: The authors have incorporated a paragraph that strongly highlights the focus in pp. 3-4, para 4:

“With a view to finding a solution to the questions posed above, this article focuses on identifying the principal data barriers existing in international techno-regulatory framework (that are followed by ship owners, classification societies as well as the operational folks engaged under the title of service providers), including the potential solutions for consideration. International norms and standards set by the International Association of Classification Societies (IACS) are the ones that regulate all RAS vessel survey activities and sporadically revised as needed to ensure that the provisions are fit-for-purpose. Based on what is found, it is correct to state that shipowners lack a techno-regulatory safety-net to protect non-personal data linked to their asset. Growing needs to eliminate liability drawbacks from such absence are being discussed in different platforms at the European Union level, including European Commission funded project ROBotics technology for Inspection of Ships (ROBINS).”

Motivation:

The authors have added five sentences in p. 3, para 3:

“The motivation behind seeking the above answers emanate from the current vacuum that exists in the regulatory setting that could call on liability implications that are purely legal in nature. It is important that service robotics engineers, as they move from “human-in-the-loop” technology to “full autonomy” technology, remain aware that technology and law reside in the same continuum and therefore, interrelated. Technology should progress, but in that process, it must be ensured that such progress does not breach or violate any contemporary legal provisions that are globally sensitive in nature. Data acquisition is one subject matter of crucial importance to both engineers as well as global, regional and national regulators. Solution is required before RAS is in mass deployment and before the topic data protection reaches the contentious stage.”

Specific comments. In general, the text is well structured and has clearly defined topics. The abstract is a good guide for what follows, even though could be refined such that it describes the motivation of the paper. More or less all fundamental theory details that are needed are discussed and concluding remarks are sufficient.

The authors have added a “motivation” sentence in p. 1 of Abstract to add strength:

“The impetus behind this study stems from the need to enquire whether “data” provisions within the realm of international techno-regulatory framework is sufficient, well-organized and harmonized so there is conflict with promulgated theoretical dimensions of data that drives all subject matter-oriented actions”.

Some comments for improvement:

1. Authors could consider use a more descriptive title for their work such that the exclusion of the “legal research” to be more obvious to the reader.

After careful consideration, the authors have revised the title and added the words “techno-regulatory” in the title so that the legal-research part derived from the international body that dictates, in principle, a topic that is predominantly technical in nature.

2. In addition, authors could consider adding a description on the number of published papers and the way for the selection of the specific etc.

The authors have now added an Appendix (at the end of the article) comprised of a tabular overview of all published literature including primary sources (in order according to year of publication) as well as the selection criteria.

The authors feel that this specific comment adds to the strength of the article and is a very constructive way forward. Reference to Appendix is provided in s. 3.4 of the article. We are extremely thankful for this very constructive comment.  

3. Do authors have information regarding the technical requirements for data processing or time consumption of algorithms? Such information would be of great importance.

We thank the reviewer for picking up on a very important point.

Having received the comments, the authors touched based with the concerned BUGWRIGHT2 Consortium Member for further information. The response was as follows:

“Regarding your question about RIT data processing computing requirements, I only have some personal experience about the time needed to analyze the photos taken from drones to develop a 3D model of the inspected region. The test was carried out in ROBINS project. Around half a day was needed. But this is not concretely confirmed. Because RIT deployment is in the “testing phase”. Then there are the three types of RIT – so the time cannot be the same for all RITs. We will know once the RITs are operated through mass deployment by several ship owners after approval from class society and flag States. I have checked for quantitative papers but they only say that the RIT process will cut back on time without substantiating the exact amount of time consumption or how fast data can be processed via algorithms. Sounds like we have another article to look forward to.”

Based on the above response, the authors were not able to provide exact information as this apparently requires further study based on deployment cases coupled with the lapse of a certain period after which data-implication cases could be observed. That said, the authors have synthesized the technical aspects in s. 3.3, para. 2. In addition, the authors have reached out to several other service providers outside the consortium, which resonates with the response shown in the previous paragraph. This certainly requires more observation and the authors look forward to emphasizing on this aspect in future publications.

4. Both Table 1 and Table 2 could be refined in order to better present relevant information.

Table 1: Table 1 has been refined with a clear division of the three RITs and information flowing under the newly created distinct headings. Color code has been introduced to help explain the different components in a befitting manner. We sincerely hope this provides a better understanding for the reader.

Table 2: Table 2 has been deleted with all information now been transferred into a new Appendix that comes under the title: Tabular Amalgamation of Published Literature & Primary Sources Based on Selection Criteria with specific reference to class society documents in the row titled “Selection Criteria: Class Society Primary Sources on Drones, Crawlers and ROV” in pp. 28 Again, we thank the reviewer for “specific comment 2” that has given the authors the idea to merge all literature and primary sources within which the class society literatures reside.

5. While authors try to explain the importance of data in maritime sector they do not mention the different difficulties due to the adverse conditions offshore. For sure this is one of the most important limitations towards the direction of Internet of Shipping (e.g. [1, [2]).

[1] Theodoropoulos, P.; Spandonidis, C.C.; Themelis, N.; Giordamlis, C.; Fassois, S. Evaluation of Different Deep-Learning Models for the Prediction of a Ship’s Propulsion Power. J. Mar. Sci. Eng. 2021, 9, 116. https://doi.org/10.3390/jmse9020116

[2] Spandonidis C.; Theodoropoulos P.; Giordamlis C. Combined multi-layered big data and responsible AI techniques for enhanced decision support in Shipping In Proceedings of The International Conference on Decision Aid Sciences and Applications 2020 (DASA20), (pp. 669-673). IEEE.

We thank the reviewer for noting this. After much consideration, we have now incorporated a sentence in s. 5, p. 25 with reference to the above sources as [85] and [86].

“Then comes other questions, such as the feasibility and effectiveness of current big data analytical techniques for increased rate and volume of data – an issue of significant importance in offshore environments that need to be considered as the industry moves towards the Internet of Shipping [85] (p. 1] [86]”.

These articles are very interesting and invoke a lot of thoughts and therefore will be considered in future publications.

6. In section 4 authors are mentioning 26 items that compromise the international regulatory blueprint. Eight of them are further discussed. A table or figure with relevant info for all items could be useful. In addition, authors could consider refining the legend of Figure 3 The source could be removed.

26 Items: The authors have summarized all 26 items from project BUGWRIGHT2 Regulatory Blueprint and incorporated all items into a new Table (Table 2, p. 20).

 Figure 3 has been refined so as to be clearer and more visible. Source has been removed.

7. Finally, since this effort lies in the borders of a review paper and a conceptual article, I would say that authors need to refine the paper, to point out the key message and the potential benefits of their work.

Again, we thank the reviewer for the comment. This has been addressed by refining the abstract, method and materials, different sections where required and the conclusion. Addressing all specific comments posed by the reviewer has been an important tool in this refinement process.

Reviewer 4 Report

The paper has been improved and the authors answered the reviewers' requests. The authors responses address all the reviewer concerns.

Some typos have to be corrected. 
The review kindly suggests rewriting the line 13-14 in a more direct way.
The reviewer kindly suggests revising the citation, i.e. in lines 62-63, different ways are used. 

In summary, this paper ends up showing important methodology. Therefore, my recommendation to the journal is to accept the manuscript.

Round 2

Reviewer 2 Report

Until now, I think JMSE is a journal specialized in Oceangraphy. Therefore, the subject of the study may be appropriate, but I think that it is appropriate to submit to a social science journal because engineering factors are not reflected in the methodology. The content and method of the paper are excellent, so if the editor is deemed suitable for the journal, it will be judged as publication.